# Functional annotations of three domestic animal genomes provide vital resources for comparative and agricultural research

Colin Kern[1], Ying Wang[1], Xiaoqin Xu [1], Zhangyuan Pan[1], Michelle Halstead [1], Ganrea Chanthavixay[1], Perot Saelao [1], Susan Waters[1], Ruidong Xiang[2,3], Amanda Chamberlain [3], Ian Korf[4], Mary E. Delany [1], Hans H. Cheng[5], Juan F. Medrano [1], Alison L. Van Eenennaam [1], Chris K. Tuggle [6], Catherine Ernst [7], Paul Flicek [8], Gerald Quon [9], Pablo Ross [1✉] & Huaijun Zhou [1✉]

Gene regulatory elements are central drivers of phenotypic variation and thus of critical importance towards understanding the genetics of complex traits. The Functional Annotation of Animal Genomes consortium was formed to collaboratively annotate the functional elements in animal genomes, starting with domesticated animals. Here we present an expansive collection of datasets from eight diverse tissues in three important agricultural species: chicken (*Gallus gallus*), pig (*Sus scrofa*), and cattle (*Bos taurus*). Comparative analysis of these datasets and those from the human and mouse Encyclopedia of DNA Elements projects reveal that a core set of regulatory elements are functionally conserved independent of divergence between species, and that tissue-specific transcription factor occupancy at regulatory elements and their predicted target genes are also conserved. These datasets represent a unique opportunity for the emerging field of comparative epigenomics, as well as the agricultural research community, including species that are globally important food resources.

[1] Department of Animal Science, University of California, Davis, Davis, CA, USA. [2] Faculty of Veterinary and Agricultural Sciences, The University of Melbourne, Melbourne, VIC, Australia. [3] Agriculture Victoria, AgriBio, Centre for AgriBioscience, Bundoora, VIC, Australia. [4] Genome Center, University of California, Davis, Davis, CA, USA. [5] USDA-ARS, Avian Disease and Oncology Laboratory, East Lansing, MI, USA. [6] Department of Animal Science, Iowa State University, Ames, IA, USA. [7] Department of Animal Science, Michigan State University, East Lansing, MI, USA. [8] European Molecular Biology Laboratory, European Bioinformatics Institute, Wellcome Genome Campus, Hinxton, Cambridge, UK. [9] Department of Molecular and Cellular Biology, University of California, David, Davis, CA, USA. ✉email: pross@ucdavis.edu; hzhou@ucdavis.edu

Genetic improvement of domestic animal species has been a key driver of reducing the environmental footprint of animal source foods, which are of critical nutritional importance in developing countries[1]. Climate change and recurring and novel pandemics, such as the current COVID-19 crisis, have unprecedented impacts on food security which, along with the ever-growing human population and increasing demand for food, mean that improvements in food production and sustainability are of critical importance. Chicken, cattle, and pig are three of the most important domestic animal species that contribute economical and nutritionally valuable protein to global food production[1]. Understanding the genetic basis of economically important complex traits in domestic animals is a primary focus of animal agriculture, as such knowledge provides the essential basis for the continued genetic improvement necessary to meet the projected increased demand using fewer animals. Furthermore, these species are important for their contributions to our understanding of evolutionary biology, human developmental biology, disease, and clinically relevant medicine[2]. It is widely accepted that most of the causative genetic variants associated with complex traits are located in non-coding genic and intergenic regions that regulate gene expression[3]. Human and mouse catalogs of regulatory elements (REs)[4–7] have been critical for identifying genetic variants associated with health and disease[8], and the recent completion of ENCODE phase 3 has further highlighted the importance of functional elements on evolutionary biology, human medicine, and genotype-to-phenotype prediction[9,10]. While some studies have investigated the evolution of regulatory sequences in non-model and non-mammalian species[11–17], broad questions still remain as to what extent the epigenomic and regulatory logic is conserved, especially at large evolutionary distances.

Here we present an eight-tissue functional annotation of the chicken, pig, and cattle genomes as one of the pilot projects of the Functional Annotation of Animal Genomes (FAANG) consortium[16,18–24]. Comparative analysis of these datasets, along with complementary datasets from the human and mouse ENCODE projects[25,26], find low levels of conservation in the sequence and position of REs, especially enhancers. On the other hand, tissue-specific patterns of transcription factor motif enrichment are highly conserved. The functional epigenetic landscape of some REs are found to be conserved across all five species, including chicken, and are associated with genes involved in basic metabolic processes. Prediction of enhancer target genes further reveal that chickens possess a reduced set of enhancers relative to mammals that collectively regulate a similar number of genes, resulting in each chicken enhancer being more multifunctional. These analyses are, to our knowledge, the largest reported genome-wide comparison of REs across birds and mammals in terms of the set of tissues and assays used, and provide a vital data resource for the agricultural research community.

## Results

**Data overview**. We performed genome-wide functional annotation using the experimental design shown in Fig. 1a. Briefly, six epigenetic data types were profiled in eight tissues (liver, lung, spleen, skeletal muscle, subcutaneous adipose, cerebellum, brain cortex, and hypothalamus) collected from sexually mature male chickens, pigs, and cattle. The epigenetic data generated included four histone modifications (H3K4me3, H3K27ac, H3K4me1, H3K27me3) and one DNA-binding protein (CTCF) using chromatin immunoprecipitation followed by sequencing (ChIP-seq)[27,28], and chromatin accessibility using DNase I hypersensitive sites sequencing (DNase-seq)[29] in chickens and

Assay for Transposase-Accessible Chromatin using sequencing (ATAC-seq)[30] in cattle and pigs. Transcriptome sequencing was also performed to correlate gene expression with regulatory region activity.

A total of 240 ChIP-seq libraries were generated and sequenced to produce 5,021,232,911 reads from chicken samples, 4,281,659,559 from pig samples, and 6,813,035,002 from cattle samples. Additionally, 15 DNase-seq libraries totaling 805,274,643 reads were produced as well as 1,038,779,370 ATAC-seq reads from 16 pig samples and 1,190,252,653 ATAC-seq reads from 15 cattle samples. The data has been deposited in public repositories (https://www.ncbi.nlm.nih.gov/geo/query/acc.cgi?acc=GSE158430) and a UCSC track hub is available to view the chromatin state prediction, predicted enhancer–gene pairs, and assay read depth (http://farm.cse.ucdavis.edu/~ckern/FAANG/).

All data generated were held to stringent data quality standards that closely mirrored the ENCODE consortium's criteria[31] (Supplementary Table 1, Supplementary Data 1 and 2). Hierarchical clustering based on the Pearson correlation of read depth in bins across the genome for the five ChIP-seq marks and the chromatin accessibility assays demonstrated data reproducibility between two biological replicates and specificity across tissues (Supplementary Figs. 2–4). The reproducibility of the RNA-seq data was similarly verified by principal component analysis (PCA) of gene expression values both within each species (Supplementary Fig. 5a) and across all three species (Supplementary Fig. 5b).

**Identification and annotation of REs**. The data generated allowed the discovery of co-occurring histone modifications, CTCF binding, chromatin accessibility, and gene expression, which was used to identify regions with regulatory function and to link them with candidate target genes. We therefore first predicted genome-wide chromatin states in each tissue within each species using ChromHMM[32] to categorize genomic regions into 14 distinct chromatin states defined by their combination of ChIP-seq marks (Fig. 1b). Labels assigned to each state were determined based on previously characterized chromatin states[33] and include active promoter and transcription start site (TSS) states, primarily defined by the presence of H3K4me3, active enhancer states with H3K27ac and H3K4me1, polycomb repressed elements marked by H3K27me3, and insulators bound by CTCF. 53%, 40%, and 31% of the chicken, pig, and cattle genomes, respectively, was annotated with a ChromHMM state corresponding to any epigenetic signal in at least one tissue, i.e., any ChromHMM state except for "Low Signal" which indicated an absence of any of the five ChIP-seq marks profiled. The percentage of the genome annotated with some regulatory function varied from tissue to tissue (Fig. 1c), reflecting the different regulatory programs responsible for tissue-specific phenotypes, as exemplified by the tissue-specific activity of the albumin (ALB) gene, which is highly expressed in the liver (Supplementary Fig. 1b).

These predicted chromatin states were then used to identify REs in each of the domestic animal genomes and annotate them with the tissues in which they were active. Next, these REs were classified as TSS proximal, genic, or intergenic based on their genomic location relative to annotated coding genes. Enrichment for each of the histone modifications assayed indicated that TSS proximal REs are characterized primarily by a strong H3K4me3 enrichment, consistent with promoter activity[34–36]. A bimodal pattern of H3K4me1 enrichment in TSS proximal REs was present, with stronger enrichment flanking the central point where the peaks of H3K4me3, H3K27ac, and chromatin

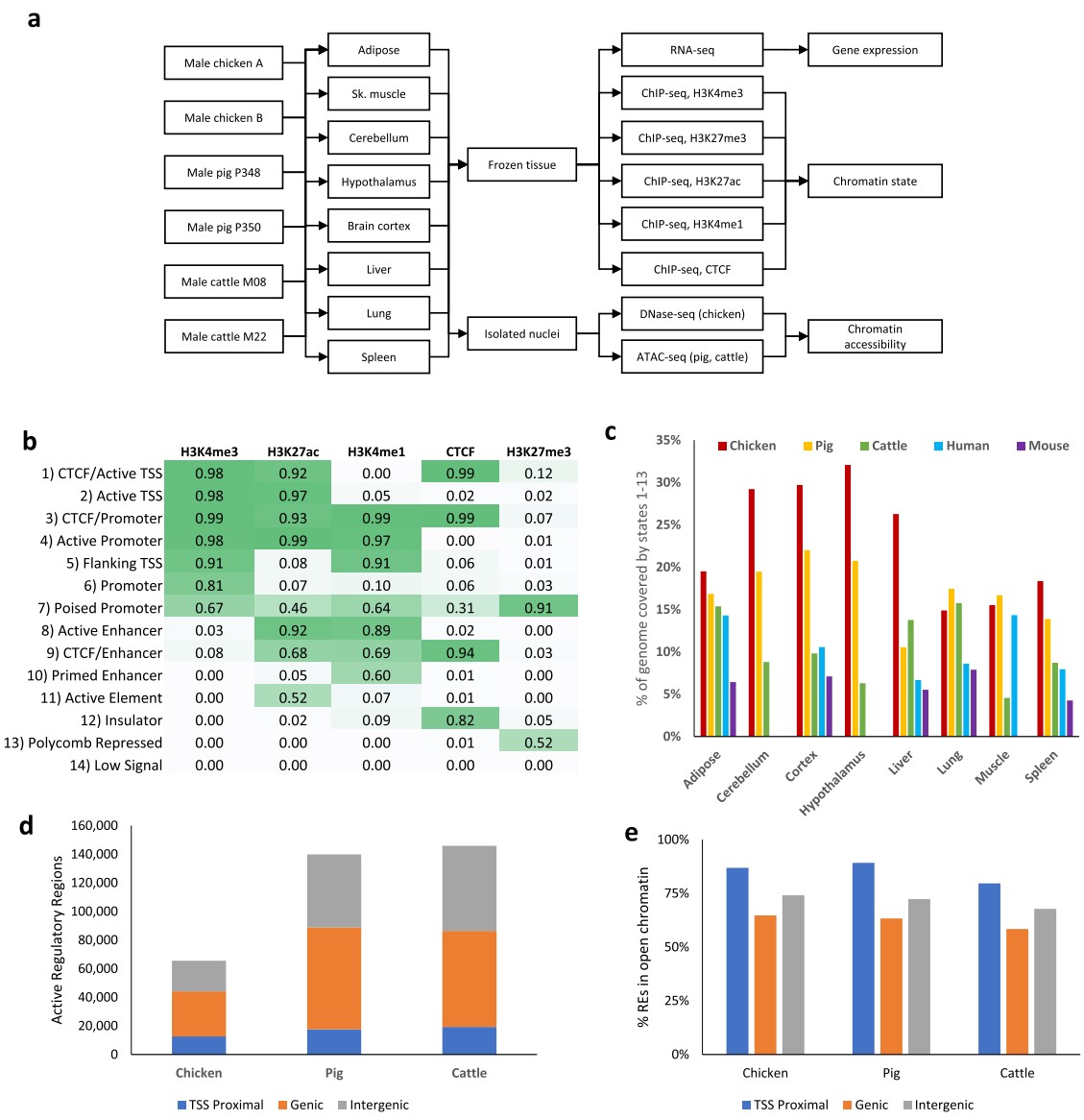

**Fig. 1 ChromHMM integrates ChIP-seq data to predict chromatin states. a** Experimental design schematic for the study. **b** Emission probabilities of the 14-state ChromHMM model. **c** Percent of the genome annotated with any functional state (any state except "Low Signal"). **d** The number of regulatory elements identified in each species, separated by TSS proximal (within 2 kb of annotated TSS), genic (overlapping annotated gene body), and intergenic. **e** Percentage of regulatory elements that co-occurred with open chromatin in the tissues in which they were active.

accessibility enrichment lie. This profile matches the enrichment of ChromHMM states around annotated TSSs, primarily with the "Active Promoter" and "Flanking TSS" states (Supplementary Fig. 1a). Genic and intergenic REs had similar profiles, with H3K27ac and H3K4me1 being the most enriched (Supplementary Fig. 1c), as is common for enhancer regions[36–38].

The number of REs identified in chickens was approximately half that found for pigs and cattle (Fig. 1d). The difference between chicken and mammals was mostly due to a lower number of genic and intergenic REs, while the number of TSS proximal REs was similar. This coincides with chickens having a similar number of genes despite the average length of gene bodies and the total size of the genome being smaller relative to mammals[39] (Chicken: 1 GB, Pig: 2.5 GB, Cattle: 2.7 GB). The majority of active REs (75±12% in chickens, 75 ± 12% in pigs, and 69±15% in cattle) were in chromatin accessible regions as determined by DNase-seq or ATAC-seq data (Fig. 1e), supporting their active function. We also observed that genic and intergenic REs had more tissue-specific activity as opposed to TSS proximal

REs (Supplementary Fig. 1d). Furthermore, of the 11,476, 12,203, and 13,074 genes expressed in chickens, pigs, and cattle, respectively (defined as TMM-normalized counts per million of at least 1), 70%, 79%, and 78% contained annotated active TSS proximal REs.

These results, taken together, revealed patterns of tissue-specificity and enrichment of histone modifications following known characteristics of promoters and enhancers. TSS proximal REs were promoter-like, as expected, while genic and intergenic REs exhibited characteristics of enhancers, with no discernible difference between the two genomic locations. Therefore, these REs are referred to as promoters, genic enhancers, and intergenic enhancers, respectively, in the following text. We then conducted comparative epigenomic analyses to explore the evolutionary conservation of REs across five species including human and mouse.

**A core set of REs is conserved across divergent amniotes.** Previous comparative studies from ENCODE and modENCODE

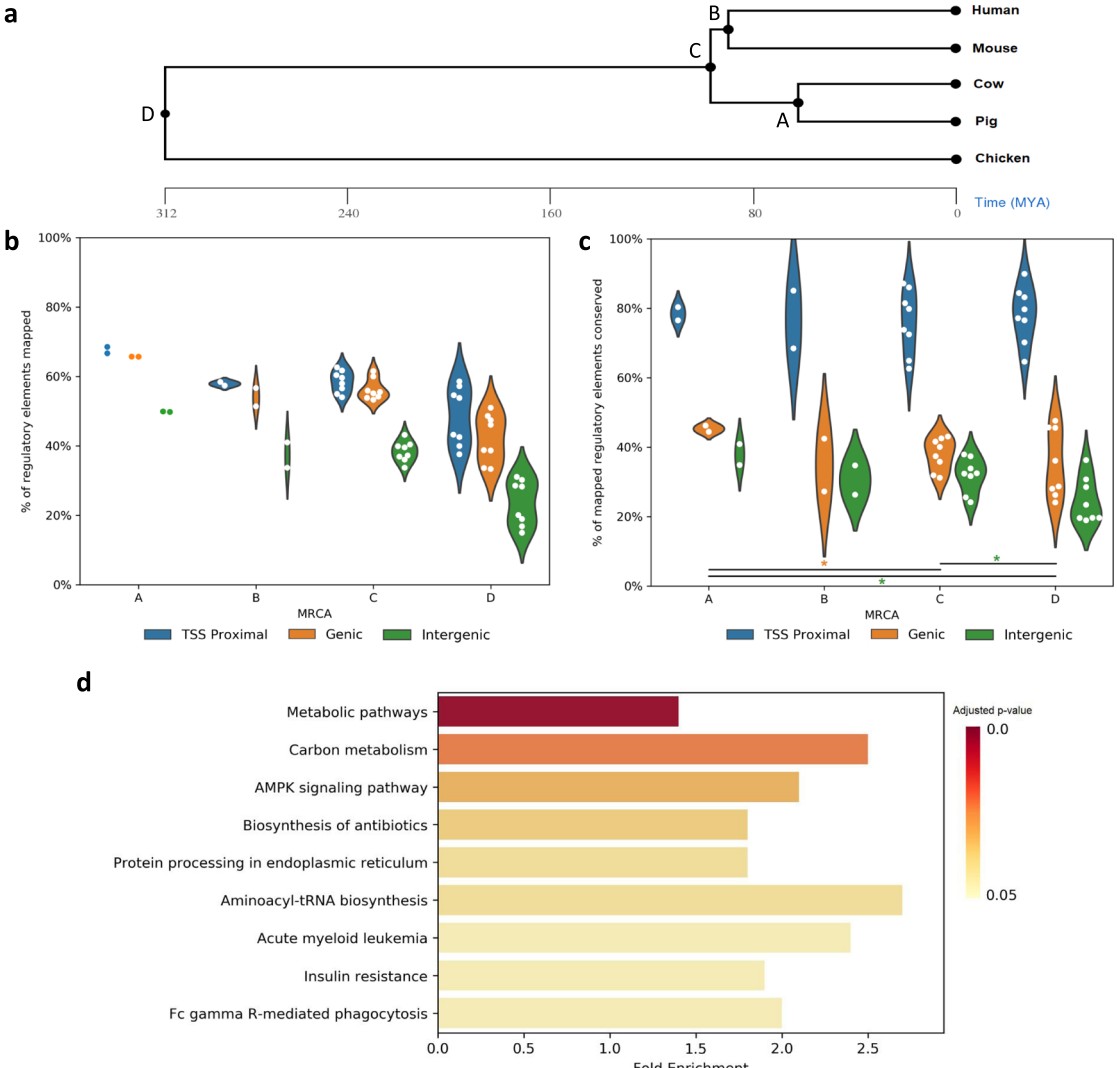

**Fig. 2 Distal regulatory elements are not positionally conserved. a** Phylogenetic tree showing the evolutionary distance between species. Most recent common ancestors (MRCA) are labeled and referenced in Fig. 3b and c. **b** Percentage of regulatory elements whose genomic coordinates could be mapped to other species, grouped by evolutionary distance. Each pair of species produces two data points, e.g. Cattle REs mapped to pig and pig REs mapped to cattle. **c** Percentage of mapped regulatory elements that were conserved, i.e. overlapped with a regulatory element identified in the target species. Asterisks indicate p-values <0.05 of a one-tailed Student's t-test (genic A–C p-value = 0.03304, intergenic A–D p-value = 0.01402, intergenic C and D p-value = 0.01238). No p-values were <0.01. **d** KEGG pathways enriched in genes with promoters conserved across all species. Benjamini–Hochberg adjusted p-values were obtained using DAVID (n = 3380 genes).

have shown that while some properties of gene regulation are highly conserved, the specific sequence and genomic position of functional REs are not[40,41]. To investigate this further with a broader selection of species, we included human and mouse along with our three domestic animal species. The coordinates of each regulatory element from each of the five species were mapped to the genomes of the other species using the Ensembl v99 alignments of amniota vertebrates. As expected, the greater the evolutionary distance between species, the lower percentage of REs mapped (Fig. 2a, b). Of particular note, intergenic enhancers had a lower mapping rate compared to promoters and genic enhancers at all evolutionary distances (Student's t-test, p-value < 0.05), while the mapping rates between promoters and genic enhancers were not significantly different at any evolutionary distance. We then checked if the mapped regulatory element from one species shared regulatory activity in the other species, indicating functional conservation of the genomic location across species. Our analysis revealed that the epigenomic landscape of mapped

promoters was conserved at an average rate of 77 ± 8% between pairs of species, while the epigenomic landscape of enhancers, including genic and intergenic, was only conserved at an average rate of 33 ± 8.1%, even though a similar proportion of promoters and genic enhancers was mapped at each inter-species comparison (Fig. 2b, c). Interestingly, the rate of epigenomic conservation for both promoters and enhancers declined at a minimal degree as evolutionary distance increased, with only one statistically significant difference in genic enhancers, between group A (conserved in pig and cattle, 45%) and C (conserved in all mammals, 38%); and two in intergenic enhancers, between group C (32%) and D (conserved in mammals and chicken, 25%) and groups A (38%) and D (25%). No significant differences were observed between groups in promoters. Taken together, our results suggested that epigenomic conservation among these five species is independent of evolutionary distance and is not always correlated with positional conservation among vertebrates including an avian species.

By examining epigenomic conservation within lineages, rather than just pairwise between species (Supplementary Fig. 6), we found a set of 9458 REs conserved across the mammals included in this study, representing similar number of promoters and enhancers. Including chicken, 3153 promoters and 1452 enhancers were conserved across all five species. This result revealed a considerable regulatory conservation across over 300 million years of evolution. For enhancers conserved across all five species, a very small number were tissue-specific, despite most enhancers being tissue-specific, suggesting these conserved enhancers are involved in basic cellular functions universal to all cell types. Further KEGG[42] pathway enrichment of genes with conserved promoters supports this notion, with the most enriched pathways related to core metabolic processes (Fig. 2d). While the sequence and position of enhancers showed low conservation, we next explored whether higher conservation exists with other features of REs such as transcription factor binding and the targeted genes they regulate.

**Tissue-specific transcription factor enrichment in active REs is highly conserved across vertebrates**. Transcription factors that bind to accessible chromatin within REs have been shown to have distinct tissue-specific activity that is conserved between mouse and human[25]. Using the chromatin accessibility data generated in this study, we performed transcription factor footprinting[43] to identify potential transcription factor (TF) binding events within characterized REs. Using these footprints, we identified 26 transcription factor motifs from the HOMER[44] vertebrate transcription factor database that were enriched in tissue-specific TF footprints in at least one tissue in each domestic animal species (Fig. 3), with the three brain tissues combined for this analysis. These transcription factor motifs showed similar patterns of enrichment across species, including human and mouse, implying a tissue-specific conserved regulatory function. FOXA2 and HNF1B, for example, were enriched and highly expressed in liver

in all three domestic animal species as well as mouse, and are known to be important for liver development[45]. The SIX1 transcription factor plays a role in adult skeletal muscle development[46] and was expressed in muscle in all three domestic animal species with motif enrichment in muscle-specific TF footprints.

**Target gene prediction of enhancers identified potential regulators conserved across species**. To predict RE target genes, we correlated gene expression across samples with the level of enrichment of histone modifications or open chromatin at enhancers. The analysis was performed on all three RE groups, as some promoters have been found to interact with other promoters in an enhancer-like manner[47]. Because this method relies on Spearman rank correlation between values across tissues, genes with small variances in expression (variance <6 CPM) were excluded from the analysis to limit false positive associations due to random chance. Similarly, REs with small variances in the enrichment of histone modifications or open chromatin were also removed as potential regulators. As it is widely recognized that enhancer–promoter interactions occur most predominantly within TADs[48], but not necessarily with the RE nearest to the gene[49–51], we predicted TADs for chickens, pigs, and cattle using CTCF-binding sites, given that Hi-C data is not available for the samples under study. Predicted TADs covered 82%, 91%, and 92% of the genomes of chicken, pig, and cattle, respectively, which is in the range of previous Hi-C data generated from mouse cell lines which identified 2200 TADs that covered 91% of the genome[47].

As a preliminary step, we measured the Spearman correlation of gene expression and ChIP-seq or chromatin accessibility signal within enhancers that overlapped or were nearest to the gene. As previously stated, REs do not always regulate their nearest gene; however, the situation is frequent enough to provide a proof-of-concept for the central assumption of our target gene prediction

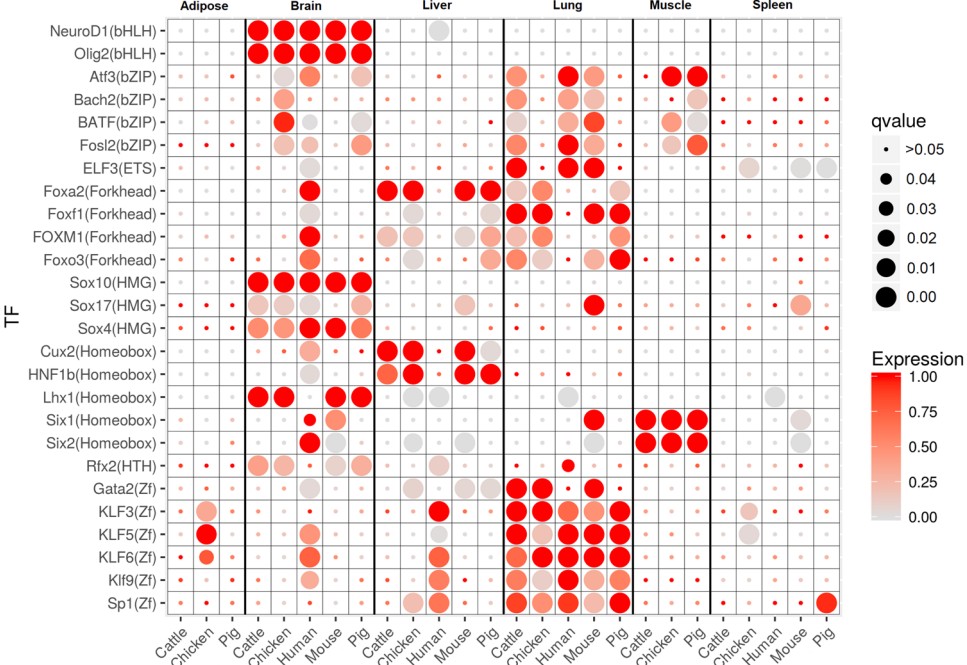

**Fig. 3 Transcription factor motifs enriched in tissue-specific footprints show similar patterns across species.** Transcription factor motifs enriched in at least one tissue in all three domestic animal species. The size of the circle indicates the statistical significance of motif enrichment (Benjamini–Hochberg adjusted p-values using HOMER), while the color indicates the expression of the corresponding transcription factor gene, normalized to the maximum expression across tissues within each species.

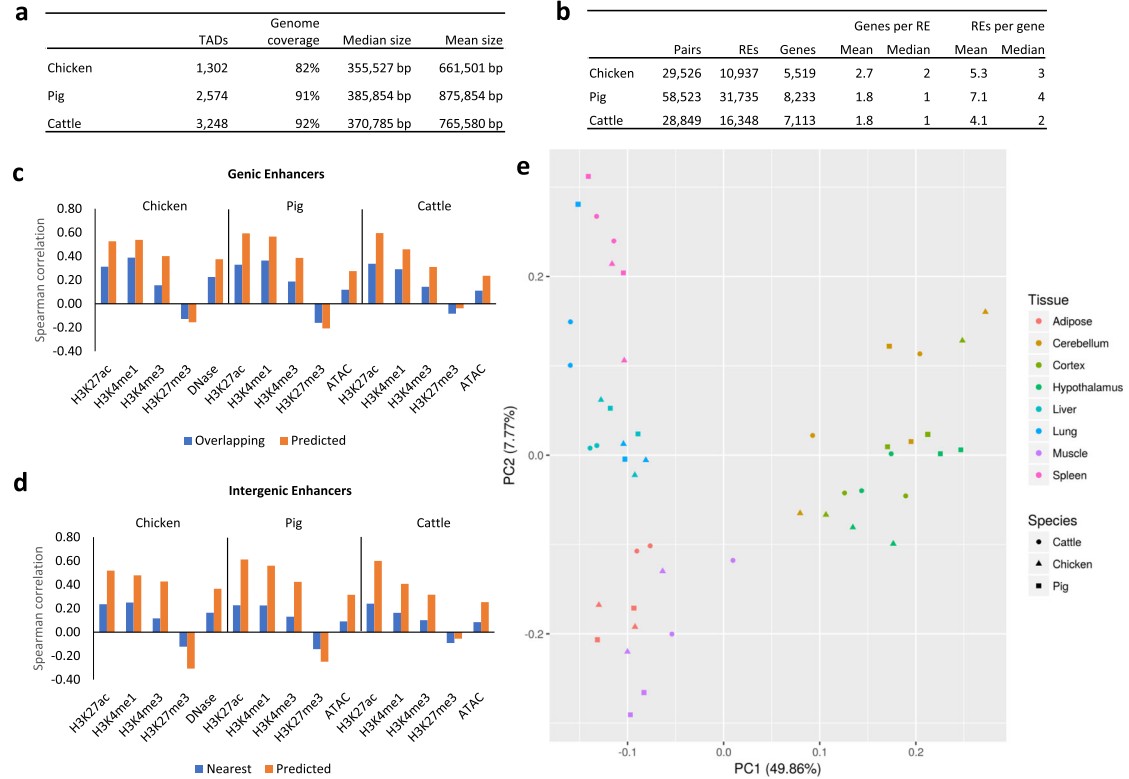

**Fig. 4 Target gene prediction of regulatory elements. a** Topologically associated domains (TADs) predicted by CTCF ChIP-seq data. **b** The number of predicted RE–gene pairs, the total number of REs and genes in at least one pair, and the mean and median number of predicted genes per RE and REs per gene. **c** Spearman rank correlation of normalized read depth in each genic RE with the expression of the gene it overlaps (blue bars) and the expression of the genes predicted as targets (orange bars). **d** Spearman rank correlation of normalized read depth in each intergenic RE with the expression of the nearest gene (blue bars) and the expression of the genes predicted as targets (orange bars). **e** Principal component plot of the normalized read depth of H3K27ac of intergenic REs predicted to target genes with one-to-one orthologs across all three species.

method—that gene expression and epigenetic signals are correlated—and to establish a baseline correlation level of these "naïve" enhancer–gene pairs. Results of this analysis indicated that H3K27ac was the most consistently correlated with gene expression at both genic and intergenic enhancers across all three species (Fig. 4c, d). Additionally, H3K27ac has been previously shown to be predictive of gene expression[52,53]. Therefore, we used H3K27ac as the signal of regulatory element activity for target gene prediction.

We predicted 29,526 RE-gene interactions in chickens (involving 10,937 REs and 5519 genes), 58,523 in pigs (31,735 REs and 8233 genes), and 28,849 in cattle (16,348 REs and 7113 genes, Fig. 4b). Most genic REs were not predicted to target the gene they overlap, with 22.1% in chickens, 35.2% in pig, and 40.4% in cattle predicted to target their overlapping gene. Because REs may have multiple predicted target genes, some genic REs that were predicted to target the gene they overlap were also predicted to target other genes, which would have not been captured with the naïve approach. In total, 92.6% of genic REs in chickens, 82.3% in pigs, and 74.6% in cattle were predicted to target a gene they do not overlap. Similarly with intergenic REs, only 14.9% in chickens, 20.7% in pigs, and 22.6% in cattle were predicted to target their nearest gene and 95.7% in chickens, 89.9% in pigs, and 87.1% in cattle were predicted to target a gene that it was not nearest to. REs in chickens were predicted to interact with more genes per RE on average compared to pigs and cattle. We verified that this was not caused by a small number of outliers with high numbers of target genes by re-calculating the average using only REs with 10 or fewer target genes. These new averages were 2.5 in chickens, 1.8 in pigs, and 1.7 in cattle. In fact, the RE with the highest number of

predicted target genes was a pig RE with 33 predicted targets, while the maximum in chickens and cattle is 23 and 22, respectively. This result suggests that chicken REs are more versatile than those of mammals. In fact, the number of RE–gene interactions predicted in chickens and cattle were very similar, despite chickens having about half the total number of REs. Compared to the previous correlations based on the nearest or overlapping gene, the Spearman correlation of gene expression with epigenetic signals of the predicted RE–gene pairs became more positively correlated with all marks, except for the repressive H3K27me3 mark, which became more negatively correlated, despite only H3K27ac being used in the prediction (Fig. 4d). This indicates that our predictions are more accurate than the naïve method of assigning enhancers to their closest gene.

To gain insight into the regulatory pathways predicted by these correlative analyses above, we first clustered REs based on their H3K27ac signal across tissues, which resulted in tissue-specific RE clusters. Next, we performed gene ontology analysis of the genes targeted by the REs in each group. These analyses revealed that REs with tissue-specific activity targeted genes with tissue-specific functions (Supplementary Fig. 7). For most clusters, the enriched GO terms show tissue-specific biological processes matching the tissues with the highest H3K27ac signal in the REs belonging to the cluster. TF motifs enriched in REs that were predicted to target genes with tissue-specific expression found numerous TFs in common across the three domestic animal species (Supplementary Fig. 8). ETS1 and FLI1, for example, were both expressed in spleen and their binding motifs enriched in REs predicted to target spleen-specific genes, suggesting a conserved tissue-specific role for these TFs.

To more directly measure the similarity of gene regulation across species, PCA on the H3K27ac enrichment values at REs predicted to target orthologous genes in each domestic animal species resulted in stronger clustering by tissue than by species in all three RE groups (Fig. 4e; Supplementary Fig. 9a, b). Taken together, these results show that while REs are not highly conserved in their genomic positions, there is tissue-specific conservation of regulatory features across species.

**An annotated data resource for comparative and complex trait analysis.** The data generated in this study represents an important resource for comparative analysis as it provides a set of epigenomic assays from the same tissues at the same developmental stage across three species with consistency in the protocols used for sample collection and data generation. As we have shown, these data show high concordance with previously reported data from the human and mouse ENCODE projects, correlate well with gene expression and chromatin accessibility, and show distinct tissue-specific patterns that relate to biologically relevant functions. Therefore, this data can be regarded a reliable epigenetic resource for these species. This dataset will facilitate further comparative epigenomic analyses, which was previously limited due to sparse epigenomic data available from species other than model organisms, as more epigenomic data is generated by the FAANG Consortium and for species outside the scope of FAANG. For researchers interested in one of the agricultural species represented by these data, the provided resources can be utilized to refine potential causative variants identified from genome-wide association studies (GWAS) for further functional validation.

As an example, 17,201,383 sequence variants associated with various complex traits in dairy cattle via expression QTL scan[54], variant function prediction[55], and GWAS were overlapped with the cattle REs identified in this study. The distribution of $p$-values showed a clear skew towards SNPs inside REs having a higher density at lower $p$-values while SNPs outside REs had higher density at higher $p$-values in traits such as milk protein content, milk fat content, and total milk volume (Fig. 5a–c). Categorizing sequence variants by types such as gene expression QTL (geQTL) or metabolite QTL (mQTL) showed that a higher percentage of these SNPs were found in REs compared to variants not in these categories (Fig. 5d). The category with the highest percentage in REs, geQTLs, appeared about 2.5 times more frequently (Fisher exact $p$-value < 0.00001) in REs compared to uncategorized SNPs, supporting the role these REs play in gene regulation. In summary, this analysis further illustrated that REs annotated in the current study can significantly narrow down the search for causative variants responsible for complex traits and fill an important gap in biology by predicting phenotype by genotype.

## Discussion

We report a large-scale analysis comparing the epigenomes, genomes, and transcriptomes of biologically diverse tissues in multiple vertebrates, including birds, and provide a comparative view of the evolutionary properties of the avian and mammalian epigenome. In general, intergenic enhancers had low genomic positional conservation compared to promoters and genic enhancers. Moreover, RE conservation across mammals and birds was independent of evolutionary distance, suggesting a core set of evolutionarily stable REs among vertebrates. Further analysis demonstrated that REs (enhancers and promoters) conserved between mammalian and avian species play essential roles in modulating genes and signaling pathways related to basic metabolic functions.

Furthermore, tissue-specific conservation of TF enrichment and target genes of RE across the vertebrate species (despite the generally low genomic level of conservation) highlight an importantfunctional role of REs in modulating biological processes. Of particular note, the number of genes regulated by each enhancer in the chicken genome were much greater than in the cattle and pig genomes. We speculate that enhancers in chickens are more multi-functional compared to their mammalian counterparts.

Finally, we demonstrated how this data can be utilized to inform studies seeking to link phenotype to genotype, such as by reducing the number of SNPs identified from a GWAS to those more likely to be causative variants. The epigenetic data and functional annotation of REs generated provide a resource for future research in animal agriculture and comparative epigenomic research. As ongoing and future FAANG projects continue, expanding the datasets to more tissues and developmental stages, as well as generating data from female individuals and exploring newer technologies such as single-cell-sequencing assays, the resource presented in this manuscript will be an important baseline upon which to compare these new datasets.

## Methods

**Genetic resources.** Animals were euthanized and tissue samples were collected with all necessary permissions granted, following Protocol for Animal Care and Use #18464, approved by the Institutional Animal Care and Use Committee (IACUC), University of California (UC), Davis. The chickens used in this study were male $F_1$ crosses of the highly inbred Line 6 and Line 7 from the USDA, ARS, Avian Disease and Oncology Laboratory (ADOL) and were euthanized with $CO_2$ under USDA inspection at 20 weeks old. Two castrated male Yorkshire littermate pigs were humanely slaughtered using electrocution in accordance with common practices in slaughterhouses at 6 months old under USDA inspection at Michigan State University. Cattle were slaughtered at UC, Davis using captive bolt under USDA inspection at 14 months old, and were intact male Line 1 Herefords that had the same sire, provided by Fort Keogh Livestock and Range Research Lab[56]. All animals were in a sexually mature adult stage when euthanized. Tissue samples were flash frozen in liquid nitrogen then stored at –80 °C until further assay processing. The tissues analyzed were chosen based on their relevance to important complex traits as well as to represent a wide range of biological functions. Two biological replicates were used per species, for a total of 16 tissue samples per species.

**Library preparation and sequencing.** The RNA-seq datasets used in this manuscript has been previously published[56]. The ATAC-seq datasets were generated using a previously published protocol[57], and is described below.

For isolation and cryopreservation of nuclei, used for DNase-seq and ATAC-seq assays, fresh tissue was minced with razor blade and transferred to gentleMACS C tube containing 10 ml of sucrose buffer, then homogenized using the gentleMACS Dissociator. Homogenate was filtered through a 100 μM Steriflip vacuum filter and DMSO added to a final 10% concentration, then pipette mixed several times and aliquoted into 2 ml cryovials. Samples were stored overnight in −80 °C in freezing container with isopropanol and then moved into storage boxes for long-term storage.

ChIP-seq experiments were performed on frozen tissue using the iDeal ChIP-seq kit for Histones (Diagenode Cat.#C01010059, Denville, NJ) according to the manufacturer's protocol except for the following changes. 20–30 mg of frozen tissue was powdered using liquid nitrogen in pre-chilled mortar. Cross-linking was performed with 1% formaldehyde which was diluted from 16% methanol-free formaldehyde (Thermo Scientific, Cat.#28906, Waltham, MA) for 8 min and quenched with glycine for 10 min. Nuclei were harvested by centrifugation at $2000 \times g$ for 5 min and resuspended in iS1 buffer for incubation on ice for 30 min. Chromatin was sheared using the Covaris E220 between 6 and 12 min depending on the tissue. For immunoprecipitation experiments, about 1000 ng of sheared chromatin (estimated from DNA extraction) was used as input after which the kit protocol was followed with 1 μg (histone modifications) or 1.5 μg (CTCF) of antibody. The following antibodies used were from Diagenode: H3K4me3 (in kit), H3K27me3 (#C15410069), H3K27ac (#C15410174), H3K4me1 (#C15410037), and CTCF (#15410210). An input (no antibody) was performed for each sample. NEBNext Ultra DNA library prep kit for Illumina libraries (New England Biolabs #E7645L, Ipswich, MA) was used for library construction, selecting for 150–200 bp (H3K4me3, H3K27ac, CTCF) or 200–400 bp (H3K27me3, H3K4me1) insert fragment sizes using Ampure beads (Beckman Coulter #A63881). Libraries were sequenced on Illumina's HiSeq 4000 with single-end 50 bp reads. A detailed protocol used to prepare the ChIP-seq libraries can be found at ftp://ftp.faang.ebi. ac.uk/ftp/protocols/assays/UCD_SOP_ChIP-Seq_Animal_tissue_20161101.pdf.

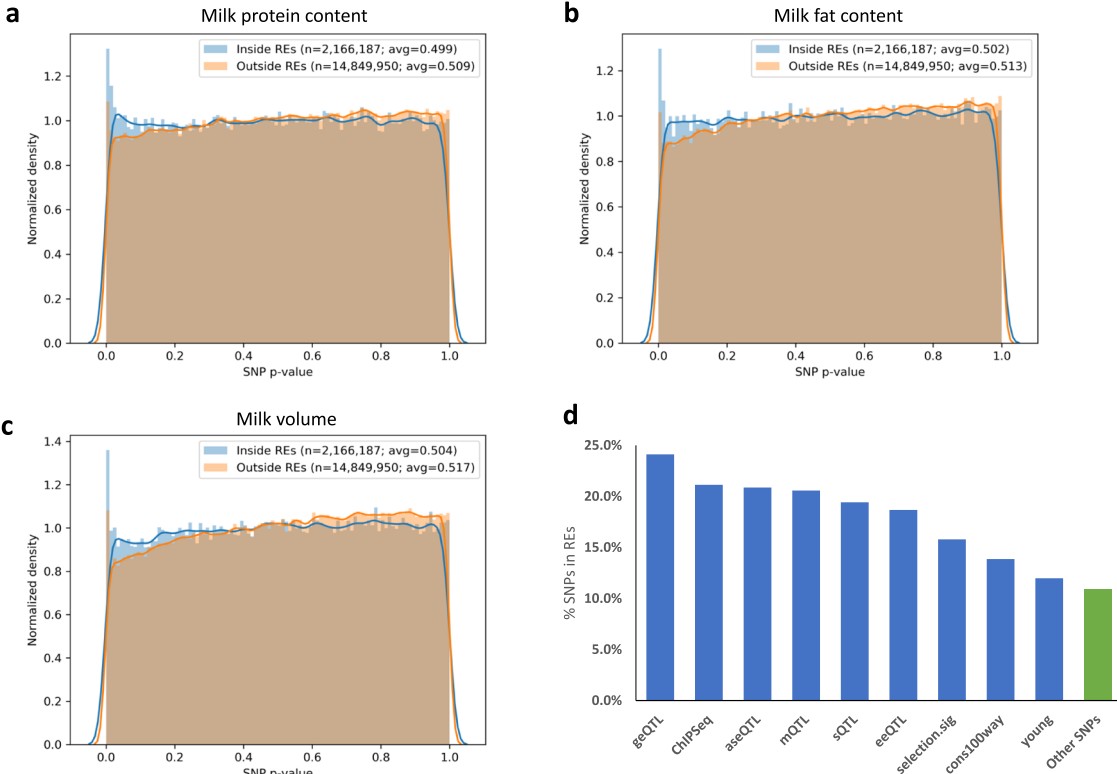

**Fig. 5 Overlap with dairy cattle GWAS SNPs. a–c** The distribution of *p*-values from GWAS for milk protein content (**a**), milk fat content (**b**), and milk volume (**c**), for SNPs inside and outside of characterized REs in cattle. The *p*-values were calculated by previous studies from which the SNPs were obtained (citations in text). **d** The percentage of SNPs in REs categorized as geQTL (gene expression QTL), ChIPSeq (SNPs in ChIP-seq peaks from previously generated H3K4me3 and H3K27ac data from liver, muscle, and mammary gland), aseQTL (allele-specific expression QTL), mQTL (metabolites QTL), sQTL (splicing QTL), eeQTL (exon expression QTL), selection.sig (selection signature between dairy and beef cattle), cons100way (variants under genomic sites conserved across 100 vertebrate species), young (variants that are recently selected), and other SNPs not placed in any of the previous categories. SNPs may belong to multiple categories.

DNase-seq datasets were generated by the Stamatoyannopoulos' Lab at University of Washington using a previously published protocol[58]. Briefly, cryopreserved nuclei were slowly defrosted on ice, pelleted at $500 \times g$, and resuspended in 37 °C DNase I digestion buffer (13.5 mM Tris–HCl pH 8.0, 87 mM NaCl, 54 mM KCl, 6 mM CaCl$_2$, 0.9 mM EDTA, 0.45 mM EGTA, 0.45 mM spermidine). After a 3 min incubation, the reaction was stopped with 2× stop buffer (50 mM Tris–HCl pH 8.0, 100 mM NaCl, 0.1% SDS, 100 mM EDTA pH 8.0, 1 mM spermidine, 0.3 mM spermine). Samples were treated with RNase for 1 h at 37 °C, then an additional hour at 55 °C with proteinase K. DNA fragments were isolated, libraries prepared, and sequenced on Illumina's Hiseq 2500 with 50 bp paired-end sequencing.

For ATAC-seq, cryopreserved nuclei prepared from fresh tissue (as described above) were slowly thawed on ice, centrifuged for 5 min at 500 rcf, and resuspended in cold PBS. Nuclei were then washed once with cold ATAC-seq cell lysis buffer and resuspended with cold PBS for counting on a hemocytometer. Approximately 200K cryopreserved nuclei were centrifuged, supernatant aspirated, then treated with 50 μl transposition mix (25 μl TD buffer, 2.5 μl TDE1, 22.5 μl ddH$_2$O) for 1 h at 37 °C and 500 rcf. DNA was purified using the MinElute PCR purification kit (Qiagen, #28004, Germantown, MD) and diluted with 10 μl EB buffer. DNA was amplified with primers whose detailed descriptions are found in Buenrostro et al. (2013)[30]. Libraries were purified using the MinElute PCR purification kit and run on Agilent Bioanalyzer (Agilent, Santa Clara, CA) for quality traces. Size-selection on libraries for 150–250 bp fragments was performed on PippinHT system (Sage Science, Beverly, MA). Libraries were sequenced on Illumina's NextSeq with PE 40bp reads.

**Genomes and annotations**. Across all analyses, the GalGal6 genome and Ensembl version 99 annotation was used for chickens, the Sscrofa11.1 genome and Ensembl version 99 annotation for pigs, and ARS-UCD1.2 genome and Ensembl version 99 annotation used for cattle.

**Processing of high-throughput sequencing data**. RNA-seq reads were trimmed with Trim Galore! 0.4.5 (https://www.bioinformatics.babraham.ac.uk/projects/trim_galore/) using default parameters, then aligned with STAR[59] 2.5.4a (--outFilterMultimapNmax 20 --alignSJoverhangMin 8 --alignSJDBoverhangMin 1 --outFilterMismatchNmax 999 --alignIntronMin 20) to the respective genome.

Alignments with an alignment MAPQ score <30 were filtered using SAMtools[60] 1.10. Gene counts were determined using htseq-count[61] 0.11.1, and then TMM normalization was performed with EdgeR[62]. Genes with a TMM-normalized counts per million (CPM) equal to or above 1 were considered expressed. Genes were considered to have tissue-specific expression if they were expressed at least 4-fold higher than all other tissues.

For broad marks (DNase-seq, ATAC-seq, and H3K27me3), a depth of 40 million aligned and filtered reads was targeted, while for the narrow marks, the target was 20 million. ChIP-seq reads were trimmed with Trim Galore! 0.4.5 using default parameters, then aligned using BWA[63] 0.7.17 with the "mem" alignment mode and default parameters. Alignments with a MAPQ score <30 were filtered using SAMtools 1.10, and then duplicates were marked and removed using the Picard toolkit[64] 2.18.17. Various quality metrics were calculated and are summarized in Supplementary Table 1, with detailed per-library statistics in Supplementary Files 1 and 2. The non-redundant fraction (NRF) is the ratio of reads after de-duplication to reads before. PCR bottleneck coefficient 1 (PBC1) is the ratio of genomic locations with exactly 1 mapped read to the total number of genomic locations with mapped reads. PBC2 is similar to PBC1, however, the denominator of the ratio is the number of genomic locations with 2 mapped reads. By ENCODE standards, an NRF in the range of 0.5–0.8 is "acceptable", a PBC1 in the same range indicates "moderate bottlenecking", and a PBC2 between 3 and 10 is labeled "mild bottlenecking". The normalized strand coefficient (NSC) and relative strand coefficient (RSC) were calculated using SPP[65] to estimate enrichment of the ChIP signal, where an NSC >1.1 and an RSC >1 indicate acceptable enrichment. The Jensen–Shannon distance (JSD) was calculated between the ChIP and input libraries using DeepTools[66] 3.3.0, providing a measure of enrichment that includes the input data, which NSC and RSC do not. The inclusion of the input read alignments in the JSD metric made it a more robust metric in discerning good data from bad by showing greater correlation with the number of peaks called and the fraction of reads in peaks (FRiP), as well as visual inspection of the data on a genome browser. FRiP measures the percentage of reads aligned to peak regions called by MACS2[67] 2.1.1 and was determined using DeepTools 3.3.0. Peaks were called with a *q*-value cutoff of 0.01 for H3K4me3, H3K27ac, H3K4me1, and CTCF. H3K27me3, ATAC-seq, and DNase-seq peaks were called in broad mark mode with a *q*-value cutoff of 0.05. To determine regions

of chromatin accessibility, peaks were called with the same parameters used to determine FRiP, but with DNase-seq and ATAC-seq using the combined alignments from both replicates.

The clustering of ChIP-seq and chromatin accessibility data was done using DeepTools 3.3.0 to get a normalized read count in 1000 bp bins across the genome, then doing hierarchical clustering with Pearson correlation as the distance metric. ChIP-seq reads were not normalized by the input reads for this clustering, and reads were extended to 200 bp. The reproducibility of the RNA-seq data was similarly verified by PCA of gene expression values within each species (Supplementary Fig. 5a) and of the expression values of 11,317 one-to-one orthologs across all three species (Supplementary Fig. 5b). Principal component (PC) 1 separated chicken from the cattle and pig, with cattle and pig samples clustering more closely by tissue than by species. A plot of PC2 and PC3 showed clustering by tissue across all species (Supplementary Fig. 5b).

**Annotation of active regulatory regions**. ChromHMM[32] 1.19 was used to train a chromatin state prediction model incorporating all ChIP-seq data from all marks, tissues, and species. Because DNase-seq data was generated for chickens while ATAC-seq data was generated for cattle and pigs, these data sets were not used in the chromatin state model. Multiple models were trained with varying numbers of states and the final 14-state model was selected to have the maximum number of states with distinct ChIP-seq mark combinations. No other parameters were changed from their defaults. We used chromatin state labels that resembled those used for the chromatin state models created by the ENCODE projects[4,5] and assigned them to states based on their combination of histone modifications and enrichment around the TSS (Fig. 1a), as well as their enrichment at various genomic elements, conserved elements, and open chromatin regions (Fig. 1b).

To consolidate and annotate the REs in each of the domestic animal genomes, we first identified all active regions for each tissue by merging consecutive regions predicted as chromatin states associated with activity (states 1–6, 8, 9, and 11) and then combined them across tissues using BEDTools[68] 2.27.1. This step helped to avoid technical issues when comparing chromatin states across tissues, such as a broader H3K27ac peak than H3K4me3 at a TSS resulting in small regions of enhancer-associated states within promoters. The REs from individual tissues that were merged to form each combined RE were used to determine its tissue activity. REs active in only a single tissue were considered tissue-specific. The active REs were then classified into groups based on their genomic location relative to annotated coding genes in the genome. Regions within 2 kb of the TSS of an annotated protein-coding transcript were classified as "TSS Proximal REs". Regions overlapping a gene body, but not within 2 kb of a TSS, were classified as "genic REs," and the remaining regions were classified as "intergenic REs". Regions that were within 2 kb of a non-coding transcript TSS or an unannotated TSS detected from RNA-seq data (from a previous analysis of the data[56]) were excluded from these groups.

The enrichment of the four histone modifications and chromatin accessibility within each RE group was done with DeepTools 3.3.0 computeMatrix with parameters "reference-point –referencePoint center -a 3000 -b 3000".

**Conservation of REs**. Human and mouse ENCODE data for the same tissues and developmental stages were downloaded from the ENCODE Consortium and were used to perform chromatin state annotation and identify REs using the same pipeline used to process the chicken, cattle, and pig data. The GRCh38 and GRCm38 genome assemblies were used with Ensembl Annotation version 99 for both. Coordinates were mapped between genomes using Ensembl Compara's amniota vertebrate multiple sequence alignment. The evolutionary distances shown in Fig. 2a were obtained from TimeTree[69]. A regulatory element was considered conserved if its mapped coordinates overlapped with a regulatory element in another species by at least 1 bp. DAVID[70] 6.8 was used to determine enriched KEGG[42] pathways.

**Transcription factor footprinting**. To identify transcription factor footprints, the HINT tool from the Regulatory Genomics Toolbox 0.12.3 was used with –atac-seq for ATAC-seq data and –dnase-seq –bias-correction for DNase-seq data, which can identify footprints from both DNase-seq[71] and ATAC-seq[72] data. DNase-seq data in chickens generated 338,547 distinct footprints across all tissues, including 32,929 containing the CTCF-binding motif. Furthermore, ChIP-seq for CTCF validated 93% of these footprints. On the other hand, ATAC-seq data in pigs and cattle generated 4,976,047 and 4,345,973 in pigs and cattle, respectively, with 45% of 89,245 CTCF footprints in pigs and 43% of 70,171 CTCF footprints in cattle validated by CTCF ChIP-seq. The difference in the number of footprints identified in mammals compared to chickens, as well as the difference in the agreement of CTCF occupied footprints with ChIP-seq, was due to a disparity in the two open chromatin assays used, rather than a biological difference between mammals and avian, as confirmed by ATAC-seq performed on the same chicken lung tissue that was used to generate the DNase-seq data for chicken lung[57] (94,376 DNase-seq footprints, 92% of 5888 CTCF footprints validated by a CTCF ChIP-seq peak; 797,042 ATAC-seq footprints, 52% of 5307 CTCF footprints with a ChIP-seq peak).

Enrichment of transcription factor motifs in footprints was done by adding a 10 bp flanking region on each side of the footprint and then using HOMER[44] 4.10 with default arguments with the exception that the given region sizes were used (default is to use 200 bp windows centered on the middle each region) and the known vertebrate motif database provided with the HOMER software was used in place of the default motif database.

**Prediction of target genes**. To remove genes with low variance in expression, the ratio of the maximum expression across samples to the minimum expression was compared to a cut-off threshold. A threshold of 6 was chosen because it removed ~3800 genes from the analysis, which is the number of housekeeping genes identified in humans by a previous study[73]. The same threshold was used to remove distal REs with low variance in their level of enrichment from the analysis.

TADs in each species were predicted using pooled CTCF ChIP-seq data from all tissues, as previous studies have suggested that while chromatin interactions within TADs may differ across cell types, the TAD boundaries themselves are stable across tissues[74] and even species[75,76]. TADs were predicted from CTCF ChIP-seq peaks using the method in Oti et al. [77]. Briefly, the CTCF peaks from all tissues were merged, then FIMO[78] was used to identify peaks containing the CTCF-binding motif. The directionality of the motif within peaks was used to match corresponding boundaries of DNA loops. Nested and overlapping loops were then merged to form the predicted TADs.

To predict RE–gene pairs, the Spearman rank correlation of every possible combination of regulatory element H3K27ac signal and gene expression value within each TAD was calculated. The gene expression value used was the TMM-normalized CPM described previously, and the H3K27ac signal was calculated by TMM-normalizing the number of H3K27ac reads aligning to each RE using the same method to normalize gene expression values. Benjamini–Hochberg adjustment was used to adjust the $p$-values for multiple testing, and adjusted $p$-values < 0.05 were considered indicative of putative interacting pairs.

**Overlap analysis of REs with SNPs from dairy cattle GWAS study**. Imputed sequence variants obtained from GWAS on dairy cattle traits from a previous study were mapped from the UMD-3.1 version of the cattle genome to the UCD-ARS1.2 version used in this paper using the UCSC liftOver tool[79] with default parameters. SNPs were then checked for their presence within REs using BEDTools[68].

**Reporting summary**. Further information on research design is available in the Nature Research Reporting Summary linked to this article.

## Data availability

Raw sequencing data and processed data has been deposited in the Gene Expression Omnibus (GEO) and is available under accession GSE158430. Accession numbers for ENCODE and Roadmap datasets used are given as Supplementary Data 3. Aligned and filtered reads, peak calls, chromatin state predictions, and identified regulatory elements are available at http://farm.cse.ucdavis.edu/~ckern/Nature_Communications_2020/. Source data are provided with this paper. Source data are provided with this paper.

## Code availability

The computational pipeline used for the analyses in this manuscript is available in GitHub[80] (https://github.com/kernco/functional-annotation).

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

## Acknowledgements
This study was funded by Agriculture and Food Research Initiative Competitive Grant nos. 2015-67015-22940 and 2018-67012-28026 from the USDA National Institute of Food and Agriculture. Additional support was provided by Pork Checkoff, Aviagen, and Multistate Research Project NRSP8 Cattle, Poultry and Swine Coordination and NC1170 (H.Z.), and the California Agricultural Experimental Station (H.Z.).

## Author contributions
C.K. performed all data analysis and wrote the manuscript. Y.W., X.X., Z.P, G.C., P.S., and S.W. worked on preparation and sequencing of ChIP-seq libraries. M.H. prepared and sequenced ATAC-seq libraries and assisted in some steps of analyzing the ATAC-seq data. R.X. and A.C. contributed the GWAS SNPs from dairy cattle. H.H.C. and C.E. provided experimental animals. I.K., M.E.D., H.H.C., J.F.M., A.L.V.E., C.K.T., C.E., P.F., G.Q., P.R., and H.Z. contributed significantly to the experimental design. P.R and H.Z. supervised the study. All authors provided feedback while drafting the manuscript and approved the final version.

## Competing interests
The authors declare no competing interests.
