## [Peer Review File · Nature Communications]

Reviewers' Comments:

Reviewer #1:

Remarks to the Author:

The authors have tried to annotate genome-wide regulatory elements in chicken, pig and cattle and conducted a large-scale analysis comparing epigenomes, genomes, and transcriptomes of eight tissues in other vertebrates. The results show that intergenic enhancers have low genomic positional conservation compared to promoters and genic enhancers, and that some regulatory elements are functionally conserved across mammals and birds despite of their evolutionary distance. The results and especially the datasets provide useful resources for future agriculture and comparative epigenomic studies on domestic animals.

Major points:

1. The Dnase-seq and ATAC-seq show significant difference. What cause this difference and how does it impact the downstream analysis? Why not to use the same technology, say, ATAC-seq which is a robust technology, to improve the integrity of the paper?
2. As the cost for Hi-C sequence is not high when you only want to call TADs nowadays. It is unusual that the authors predicted TAD regions just by the CTCF chip-seq data, which is not a common method. Furthermore, the cited article Oti, et al.⁷⁶ where the prediction method comes from is a method to predict CTCF loops rather than TAD. More importantly the TAD data of the three species have been published. You could download the data to confirm your results. Moreover, you could compare the TAD results predicted by CTCF with the published TAD data for each species. Overall, more analyses and explanations are needed to demonstrate this problem.
3. Usually chicken can not be called livestock, and it belongs to fowl. "Domestic animals" should be the right term.

Minor points:

1. The figures in article look boring. There are too many histograms in main article and the colors are not coordinated. The authors may find more way to visualize their data efficiently and beautifully.
2. There are some symbol errors in some places. For example, "Trim Galore!" should be "Trim Galore".
3. Some parameters of used softwares are not described. For example, the parameters of ChromHMM.
4. Line 217. The two – shouldn't be here.
5. Line 335. There are some typos in some places. For example, "iDeal" should be "ideal".
6. Line 445. The explanation of the name should be at where it first appears. For example, transcription start site (TSS) should not be here.
7. Line 445. Why the cutoff of TSS is 2kb? Have you tested other cutoffs, like 3kb or 1kb?

Reviewer #2:

Remarks to the Author:

Comments to Authors

Overall, the paper is well-written, and the resources newly generated here (i.e., functional genome annotations of three farm animal species) are of general interest and high impact in the wider fields including animal/human genetics and comparative genomics. The statistical methods used are solid. Through comparing epigenomes of multiple tissues across five species (including human and mouse), Dr. Zhou and his colleagues characterized the conservation of regulatory elements, particularly for enhancers. Taking cattle as example, they also showed the importance of regulatory elements in the interpretation of genetics of complex traits in livestock. I have a few major comments as shown below:

- 1) I think it could be useful to discuss the limitations of current resources and next steps. For instance, in this pilot project of FAANG, the authors only considered adult male animals. Any plans to generate tissues in females in the future, as the regulatory elements are tissue-dependent and most

traits of economic traits may be relevant with female tissues, such as milk production (mammary gland) and egg production (ovary)? How about different developmental stages, and single-cell levels (as the difference across species may due to differences in cell composition of tissues)?

2) Could the authors biologically explain why each enhancer regulate more genes in chicken than in cattle and pig? Any technical bias from different qualities of reference genomes? Did you restrict your comparisons within 1-to-1 orthologues genes in Fig. 4b?

Minors:

1. Line 132-133: How did you define expressed genes? What about the non-expressed genes in terms of active TSS proximal REs?
2. Line 162-163: How did you define the four groups, i.e., A, B, C and D? It could be good to explain a bit in the main text as well.
3. Line 176: How about the function of genes targeted by conserved enhancers? Did they have the similar targets across species? Did they show the similar functions as genes with conserved promoters?
4. Line 223-225: Could the authors show the average distances (or the distribution of distances) between RE and their predicted targets? What proportion are they overlapped with the "naive" approach (i.e., nearest genes)?
5. Line 225-226: The authors found that each RE interact with more genes in chicken than in pig and cattle. Is this due to outliers (i.e., a few enhancers interact with many genes in chicken)?
6. The authors found that the majority of conserved enhancers were universal to all tissues (Line 175), while they also found that there is tissue-specific conservation of regulatory features across species (Line 250-251). Did this indicate that the degree of activity (i.e., signal density of H3K27ac) of conserved enhancers is an important driver of tissue-specificity that was conserved across species?
7. Line 256-263: These were data summary. It could be better to put them in the beginning of Results section (Data Overview).
8. Line 282: Is this 2.5 times difference statistically significant?
9. Could you explain a bit why the chromatin states annotated in chicken and pig covered much larger parentages of genomes than cattle did (even human and mouse) across most of tissues (Fig. 1c)? However, cattle has more active regulatory regions than chicken (Fig. 1d).
10. Fig. 2a, Is this calculated based on the entire genome sequence information (or regions of regulatory elements)?
11. Fig. 3, could you show how did you get q-values (FDR?) in the legend?
12. Fig. 4a, Was the median/mean size measured by bp?

We are very grateful to the editor and reviewers for their efforts in reviewing our manuscript. These comments are extremely helpful in improving the strength and clarity of this paper. Below, please find a point-by-point response to each comment from the reviewers.

Reviewer #1 (Remarks to the Author):

The authors have tried to annotate genome-wide regulatory elements in chicken, pig and cattle and conducted a large-scale analysis comparing epigenomes, genomes, and transcriptomes of eight tissues in other vertebrates. The results show that intergenic enhancers have low genomic positional conservation compared to promoters and genic enhancers, and that some regulatory elements are functionally conserved across mammals and birds despite of their evolutionary distance. The results and especially the datasets provide useful resources for future agriculture and comparative epigenomic studies on domestic animals.

Major points:

1. The DNase-seq and ATAC-seq show significant difference. What cause this difference and how does it impact the downstream analysis? Why not to use the same technology, say, ATAC-seq which is a robust technology, to improve the integrity of the paper?

Response: The reviewer brought up a great point. The DNase-seq and ATAC-seq datasets are quite similar in terms of the identification of open chromatin regions in the genome, as can be seen in Fig 1e with a similar percentage of regulatory elements in open chromatin regions across all three species. We have also shown in a previous publication that open chromatin regions measured from DNase-seq and ATAC-seq on the same tissue (chicken lung) overlap substantially (<https://doi.org/10.1038/s41598-020-61678-9>). ATAC-seq data produces a higher number of peaks than DNase-seq, but the genome coverage of these peaks is similar because ATAC-seq tends to produce more narrow peaks. When it comes to identifying transcription factor occupancy with the footprinting method, a significant difference is indeed present. The HINT tool from the Regulatory Genomics Toolkit that we used has made significant progress in accounting for the biases present in these assays; however, there is still a difference between the two assays. We believe the DNase-seq footprinting results to be the most accurate, while the ATAC-seq footprinting results have a higher false positive rate. As these results were used for the transcription factor enrichment analyses, the higher false positive of footprints in ATAC-seq would have the effect of making it harder to achieve significant TF enrichment. Nevertheless, we are still able to show similarities in enrichment across species. It is possible that DNase-seq data from all species would have shown more similarities; however, the similarities found with ATAC-seq data we believe are real. The original experimental design for this project was to use DNase-seq in all species by Prof. John Stam's Lab at University of Washington; however, after data was generated for chicken, the difficulty, budget, and time required to produce high quality data made it unrealistic to produce such datasets for pig and cattle. With ATAC-seq being now the most commonly used method to map open chromatin we switch to this assay for pig and cattle.

2. As the cost for Hi-C sequence is not high when you only want to call TADs nowadays. It is unusual that the authors predicted TAD regions just by the CTCF chip-seq data, which is not a common method. Furthermore, the cited article Oti, et al.⁷⁶ where the prediction method comes from is a method to predict CTCF loops rather than TAD. More importantly the TAD data of the three species have been

published. You could download the data to confirm your results. Moreover, you could compare the TAD results predicted by CTCF with the published TAD data for each species. Overall, more analyses and explanations are needed to demonstrate this problem.

Response: The reviewer brings up a valid point. We have access to Hi-C data from chicken and pig generated by another FAANG pilot project, but data from cattle is not of sufficient quality. To present the most consistent analysis across all 3 species evaluated, we decided to use the CTCF data we generated from the same identical samples used for other reported assays, rather than Hi-C data from one source for chicken and pig, and a different source from cattle. Additionally, as this paper is focused on presenting these datasets as a resource, we preferred to use our data whenever possible. The method we used from Oti et al. was employed to predict CTCF-mediated loops in from all tissues, which were then merged into TADs. We have clarified this in the Methods section on lines 530-531.

3. Usually chicken can not be called livestock, and it belongs to fowl. "Domestic animals" should be the right term.

Response: We have adjusted the terminology used in our manuscript.

Minor points:

1. The figures in article look boring. There are too many histograms in main article and the colors are not coordinated. The authors may find more way to visualize their data efficiently and beautifully.

Response: It is not clear what the reviewer means by "the colors are not coordinated". We have chosen to use a different color scheme for the RE types (blue, orange, gray) and for the species (red, yellow, green), to make it easier to distinguish what is being illustrated in each panel. We have double-checked the figures and make sure they are consistent with this coloring scheme throughout the manuscript.

2. There are some symbol errors in some places. For example, "Trim Galore!" should be "Trim Galore".

Response: The exclamation mark is part of the official name of the tool, and so we have left it as is.

3. Some parameters of used softwares are not described. For example, the parameters of ChromHMM.

Response: We have added more details to the methods section for the parameters used for the analysis tools.

4. Line 217. The two – shouldn't be here.

Response: We do not see an instance of "the two" in or around line 217, so are not sure what the reviewer is referring to. The nearest instances appear to be on line 174 and 339.

5. Line 335. There are some typos in some places. For example, "iDeal" should be "ideal".

Response: We appreciate the reviewer's careful review. This is not a typo and has been left as is. "iDeal" is the brand name of a reagent kit.

6. Line 445. The explanation of the name should be at where it first appears. For example, transcription start site (TSS) should not be here.

Response: Thank you for spotting this error. We have checked that various abbreviations are defined at their first appearance and made corrections.

7. Line 445. Why the cutoff of TSS is 2kb? Have you tested other cutoffs, like 3kb or 1kb?

Response: The 2 kb cutoff is the most widely used range for defining promoter regions based on gene annotations and was used to be consistent with previous epigenomic studies. For example, this paper on the development of ChromHMM using ENCODE data explored “promoter-associated states” by looking for enrichment within a 2 kb window of TSSs (<https://doi.org/10.1038/nbt.1662>).

Reviewer #2 (Remarks to the Author):

Comments to Authors

Overall, the paper is well-written, and the resources newly generated here (i.e., functional genome annotations of three farm animal species) are of general interest and high impact in the wider fields including animal/human genetics and comparative genomics. The statistical methods used are solid. Through comparing epigenomes of multiple tissues across five species (including human and mouse), Dr. Zhou and his colleagues characterized the conservation of regulatory elements, particularly for enhancers. Taking cattle as example, they also showed the importance of regulatory elements in the interpretation of genetics of complex traits in livestock. I have a few major comments as shown below: 1) I think it could be useful to discuss the limitations of current resources and next steps. For instance, in this pilot project of FAANG, the authors only considered adult male animals. Any plans to generate tissues in females in the future, as the regulatory elements are tissue-dependent and most traits of economic traits may be relevant with female tissues, such as milk production (mammary gland) and egg production (ovary)? How about different developmental stages, and single-cell levels (as the difference across species may due to differences in cell composition of tissues)?

Response: The reviewer’s points are well-taken! We have included a mention of the next steps and future efforts in the Conclusions section. See lines 327-331.

2) Could the authors biologically explain why each enhancer regulate more genes in chicken than in cattle and pig? Any technical bias from different qualities of reference genomes? Did you restrict your comparisons within 1-to-1 orthologues genes in Fig. 4b?

Response: While we present some speculation in the manuscript, testing of hypotheses to answer this question was beyond the scope of this study. In terms of reference genome quality, the chicken and cattle are similar, while pig has a more fragmented assembly. The comparisons were not restricted to 1-to-1 orthologs.

Minors:

1. Line 132-133: How did you define expressed genes? What about the non-expressed genes in terms of active TSS proximal REs?

Response: We defined expressed genes as those with a TMM-normalized counts per million over 1. This is specified in the Methods section, but we have added this information on line 143 as well.

2. Line 162-163: How did you define the four groups, i.e., A, B, C and D? It could be good to explain a bit in the main text as well.

Response: We have added descriptions of these group to the main text on lines 173-175.

3. Line 176: How about the function of genes targeted by conserved enhancers? Did they have the similar targets across species? Did they show the similar functions as genes with conserved promoters?

Response: The target gene prediction presented in the manuscript relies on comparing patterns of variation in gene expression and histone modification enrichment across the tissues analyzed, so it is unable to predict enhancer-gene interactions involving genes with consistent expression across tissues. The conserved enhancers for the most part do not have predicted targets, which is consistent with the conserved promoters that were located at genes involved in universal metabolic processes.

4. Line 223-225: Could the authors show the average distances (or the distribution of distances) between RE and their predicted targets? What proportion are they overlapped with the “naive” approach (i.e., nearest genes)?

Response: We have added this analysis to the text on lines 237-245.

5. Line 225-226: The authors found that each RE interact with more genes in chicken than in pig and cattle. Is this due to outliers (i.e., a few enhancers interact with many genes in chicken)?

Response: The reviewer raised a great point! The median value is also higher in chicken, indicating that the higher mean is not only due to outliers. To verify, we recalculated the statistics using only REs with 10 or fewer target genes. The new values are 2.5 in chickens, 1.8 in pigs, and 1.7 in cattle. In fact, the RE with the highest number of predicted target genes is a pig RE with 33 predicted targets. The maximum in chickens is 23 and in cattle it is 22. We have added this to the text on lines 246-251.

6. The authors found that the majority of conserved enhancers were universal to all tissues (Line 175), while they also found that there is tissue-specific conservation of regulatory features across species (Line 250-251). Did this indicate that the degree of activity (i.e., signal density of H3K27ac) of conserved enhancers is an important driver of tissue-specificity that was conserved across species?

Response: While tissue-specific enhancers are indeed associated with tissue-specific activity as measured by H3K27ac, further study would be required to say whether this is a causative relationship and its importance in comparison with other possible factors. Here, we are simply using H3K27ac as a measure of activity to identify the level of activity of each RE in each tissue.

7. Line 256-263: These were data summary. It could be better to put them in the beginning of Results section (Data Overview).

Response: We have moved these sentences to the beginning of the Results section at lines 88-95.

8. Line 282: Is this 2.5 times difference statistically significant?

Response: Yes, it is significant based on a Fisher exact test. We have added this to the text.

9. Could you explain a bit why the chromatin states annotated in chicken and pig covered much larger parentages of genomes than cattle did (even human and mouse) across most of tissues (Fig. 1c)? However, cattle has more active regulatory regions than chicken (Fig. 1d).

Response: The reviewer had a great observation! There are a few factors in play here. First, the percentage of the genome annotated with chromatin states (Fig 1c) includes states such as polycomb repressed and primed enhancers, which were not used to define active regulatory regions. It should also be noted that less coverage of a region could result in more numerous, but smaller, REs compared to another similar-sized region with more coverage which would have fewer but larger REs. The complexity of the histone modification combinations make it difficult to extract discrete regulatory elements in a way that is consistent across all the different chromatin state landscapes that result from this type of analysis.

In reference to the lower genome coverage of the mouse and human, this is most likely due to lower sequencing depth in the ENCODE data, which was generated at a time when their guidelines recommended roughly half the depth that we targeted for this manuscript (and that current ENCODE guidelines now recommend).

10. Fig. 2a, Is this calculated based on the entire genome sequence information (or regions of regulatory elements)?

Response: The phylogenetic tree in this figure is provided as a reference for readers unfamiliar with the evolutionary relationship of the species in this study, and to aid in visualizing the A, B, C, and D groups of this analysis. The evolutionary distances are obtained from the TimeTree database and not calculated from our data. We have added this information and cited the database in the Methods section.

11. Fig. 3, could you show how did you get q-values (FDR?) in the legend?

Response: The q-values are provided by the Homer motif enrichment tool and calculated from the Benjamini-Hochberg procedure. We have added this to the figure caption.

12. Fig. 4a, Was the median/mean size measured by bp?

Response: Yes, these sizes are in bp, which is now indicated in the figure.

Reviewers' Comments:

Reviewer #1:

Remarks to the Author:

The revision has largely addressed my concerns, and I have no more question.

Reviewer #2:

Remarks to the Author:

Thanks for the revising, and my concerns have be addressed. I do not have further comments on the manuscript.